# Associated Anomalies and Outcome in Patients with Prenatal Diagnosis of Aortic Arch Anomalies as Aberrant Right Subclavian Artery, Right Aortic Arch and Double Aortic Arch

**DOI:** 10.3390/diagnostics14030238

**Published:** 2024-01-23

**Authors:** Roland Axt-Fliedner, Asia Nazar, Ivonne Bedei, Johanna Schenk, Maleen Reitz, Stefan Rupp, Christian Jux, Aline Wolter

**Affiliations:** 1Division of Prenatal Medicine, Department of Obstetrics and Gynecology, Justus-Liebig-University, Giessen and University Hospital, Giessen & Marburg, 35392 Giessen, Germany; roland.axt-fliedner@gyn.med.uni-giessen.de (R.A.-F.); nazar_a@ukw.de (A.N.); ivonne.bedei@gyn.med.uni-giessen.de (I.B.); johanna.schenk@gyn.med.uni-giessen.de (J.S.); maleen.reitz@gyn.med.uni-giessen.de (M.R.); 2Department of Paediatric Cardiology, Children’s Hospital, Justus-Liebig-University, Giessen and University Hospital, Giessen & Marburg, 35392 Giessen, Germany; stefan.rupp@paediat.med.uni-giessen.de (S.R.); christian.jux@paediat.med.uni-giessen.de (C.J.)

**Keywords:** aortic arch anomalies, aberrant right subclavian artery (ARSA), right aortic arch with mirror image branching, right aortic arch with aberrant left subclavian artery (RAA-ALSA), double aortic arch (DAA), vascular ring/sling

## Abstract

We aimed to evaluate retrospectively associated anomalies and outcome in prenatal aortic arch anomalies (AAAs). We included ninety patients with aberrant right subclavian artery (ARSA), right aortic arch (RAA) with mirror image branching (RAA-mirror) or aberrant left subclavian artery (RAA-ALSA) and double aortic arch (DAA) between 2011 and 2020. In total, 19/90 (21.1%) had chromosomal anomalies, the highest rate being within the ARSA subgroup (17/46, 37%). All (13/13) of the RAA-mirror subgroup, 10/27 (37.0%) of RAA-ALSA, 13/46 (28.3%) of ARSA and 0/4 within the DAA subgroup had additional intracardiac anomaly. The rate of extracardiac anomalies was 30.7% in RAA-mirror, 28.3% in ARSA, 25.0% in DAA and 22.2% in the RAA-ALSA subgroup. A total of 42/90 (46.7%) had isolated AAAs: three (7.1%) with chromosomal anomalies, all trisomy 21 (3/26, 11.5%) within the ARSA subgroup. Out of 90, 19 (21.1%) were lost to follow-up (FU). Two (2.2%) intrauterine deaths occurred, and six (6.7%) with chromosomal anomalies terminated their pregnancy. In total, 63 (70.0%) were liveborn, 3/63 (4.8%) with severe comorbidity had compassionate care and 3/60 (5.0%) were lost to FU. The survival rate in the intention-to-treat cohort was 53/57 (93%). Forty-one (77.4%) presented with vascular ring/sling, two (4.9%) with RAA-ALSA developed symptoms and one (2.4%) needed an operation. We conclude that intervention due to vascular ring is rarely necessary. NIPT could be useful in isolated ARSA cases without higher a priori risk for trisomy 21 and after exclusion of other anomalies.

## 1. Introduction

Aortic arch anomalies (AAAs) are caused by either abnormal persistence or abnormal regression of single or multiple embryological vascular segments of the aortic arch [1,2,3,4]. They present a large spectrum of variations and anomalies. The most common type is left aortic arch with aberrant right subclavian artery (ARSA) followed by right aortic arch anomalies with aberrant left subclavian artery. Right aortic arch with mirror image branching (RAA-mirror) and double aortic arch (DAA) are less frequent [5,6,7,8]. The frequency for AAAs is low with a prevalence of 0.7% for ARSA [9] and 0.1% for RAA in a postnatal cohort and in a prenatal cohort of low-risk pregnancies [10,11].

Although most AAAs are asymptomatic during postnatal life, prenatally they could be a marker for chromosomal anomalies; therefore, a detailed scan for further anomalies is necessary and should be considered in prenatal family counseling [12].

AAAs are associated with genetic and further cardiac and extracardiac anomalies. In the postnatal period, ARSA is present in 3% of children and adults with a congenital heart defect versus in 0.1% without a heart defect [13]. After postnatal radiographic studies, ARSA occurs in cases with trisomy 21 in up to 35% [14,15]. Chaoui et al. suggested ARSA as an ultrasound marker for trisomy 21 after preliminary studies. In a prenatal cohort with trisomy 21, they found ARSA in 35.7% [16]. Data concerning the incidence of ARSA in low-risk fetuses are rare. Zalel et al. reported an incidence of 1.4% of ARSA in a fetal cohort with routine sonographic examination [17].

The prevalence of the rare condition of fetal double aortic arch is not well known [18]. Some estimate an incidence of approximately 0.005-0.007% [19]. In isolated cases, most neonates are asymptomatic after birth and may even remain undetected. However, in rare cases they can be symptomatic such that the trachea and esophagus are completely encircled by vascular tissue and can cause difficulty in breathing or swallowing and lead to potentially fatal complications [20,21,22,23]. AAAs can be prenatally diagnosed by ultrasound visualizing of the three vessel and trachea view in the axial plane in combination with the subclavian artery view for accurate identification of an aortic arch branching pattern [9,12,16]. In the German “Mutterschaftsrichtlinien”, only four chamber view scanning is included. Prenatal detection of these anomalies however is important as aortic arch anomalies are associated with congenital heart defects and chromosomal anomalies including trisomy 21 and microdeletion 22q11.2 [16,24]. The aim of our retrospective evaluation was to evaluate associated anomalies and outcomes in prenatal cases of different AAAs.

## 2. Materials and Methods

The initial search in our fetal database retrieved 100 cases with a prenatal diagnosis of AAAs between 2011 and 12/2020 in our division of prenatal medicine at Division of Prenatal Medicine Justus-Liebig-University and University Hospital in Giessen.. In our center, a rather high-risk population presents with a low number of cases for general screening and most cases with referral for detailed fetal echocardiography.

We divided AAAs into four subgroups: ARSA, RAA with mirror branching, RAA with ALSA or DAA.

Examples of the most common forms of aortic arch anomalies (ARSA, RAA and DAA) are presented in Figure 1, Figure 2, Figure 3 and Figure 4. Neonates without postnatal confirmation of a prenatal diagnosis of AAAs were excluded from the analysis. Data were collected retrospectively from medical files, ultrasound images and videos and, if available, MRI reports. After birth, an MRI was performed in patients with additional cardiac defects and/or symptoms for vascular ring/sling. Informed parental consent to anonymized analysis of the data was obtained. Fetal echocardiography was performed according to the guidelines of ISUOG by a segmental approach and defined anatomical planes with color pulsed wave Doppler by a certificated sonographer (minimum DEGUM II qualification) including a three vessel and trachea view as well as subclavian artery views. Philips Epiq7 and Ge Voluson E10 with a 5 MHz, 7.5 MHz or 9 MHz sector or curved array probes were used. ARSA was diagnosed when the aortic arch was located on the left side and right subclavian artery arises as the last vessel from the aortic arch and crosses the fetal chest behind the trachea to the right side (Figure 1a,b). RAA-mirror was diagnosed when the branching pattern is exactly opposite to the left artic arch with normal branching patterns which means that aortic arch and ductus arteriosus were located on the right side of the trachea and the left subclavian artery was the first branch to leave the aortic arch (Figure 3). RAA with aberrant left subclavian artery (RAA-ALSA) was diagnosed when the aortic arch located at the right sight of the trachea and the left ductus arteriosus formed a U-shape confluence and the left subclavian artery arising as the last branch from the aortic arch (Figure 2a–d). DAA was diagnosed in the presence of two aortic arches on each side of the trachea with common branching arteries (Figure 4a–c). Invasive diagnostics including fetal karyotyping, fluorescent in situ hybridization for microdeletion 22q11.2 and, depending on the presence of further sonographic anomalies, also array-CGH was offered. Parental counseling by pediatric cardiologists was part of the prenatal work-up. For the data analysis, descriptive statistics was appropriate and performed using Microsoft^®^ Excel^®^ for Microsoft 365 MSO (Version 2210 Build 16.0.15726.20188). Categorial data are expressed as frequencies and percentages (%) and continuous variables are reported as mean ± SD or median (range) depending on the data distribution.

## 3. Results

The initial search retrieved 100 fetuses with a prenatal diagnosis of AAAs. Ten patients with prenatal suspicion of AAAs were excluded from analysis because the AAAs were not confirmed after birth. Most of these patients (6/10 = 60%) had prenatal suspicion of ARSA, one had TOF and suspicion of RAA-mirror and in three RAA-ALSA was assumed prenatally. Ninety patients were included in our final analysis. The median maternal age was 34 years (range 20–44). The rate of assisted reproductive technology was 9 of 90 (10.0%). In total, 8 of 90 (8.9%) cases were twin pregnancies: three monochorionic twin pregnancies and five dichorionic twin pregnancies with, in each case, one affected fetus. Patients were referred to our center at a median of 22.1 wks (11.43–30.43). In total, 15/90 (16.7%) presented within the first trimester. In one case, a prenatal diagnosis of AAA differed from a postnatal diagnosis: DAA was diagnosed during the prenatal period, but postnatal diagnostic imaging revealed RAA-ALSA. For the data analysis, this patient was encountered in the correct subgroup of RAA-ALSA. In all other cases, a prenatal diagnosis of AAAs was correct. ARSA was the most common AAA followed by RAA-ALSA, RAA-mirror and DAA as a rare form of AAA.

### 3.1. Chromosomal Anomalies in AAAs

Of the whole cohort, 19/90 (21.1%) had additional chromosomal anomalies; most (17/19 = 89.5%) patients were within the ARSA subgroup (Figure 5, red boxes). Within this subgroup, 21/46 (45.7%) had invasive prenatal diagnostics, the rate of chromosomal anomalies was 37.0% (17/46) and the overall highest incidence was observed for trisomy 21 (7/19 = 36.8%). The remaining two patients with chromosomal anomalies were within the RAA-ALSA (Figure 5, orange box) and RAA-mirror subgroups (Figure 5, green box) and had additional extracardiac and intracardiac anomalies. One had microdeletion 22q11.2 and microcephaly; in the other patient with TOF and extracardiac anomalies (microtia, glaucoma, nevus flammeus, supratentorial angioma), genetic testing revealed a GNAQ mutation. An overview of the distribution of genetic anomalies within the different subgroups of AAAs is given in Figure 5.

### 3.2. Associated Intracardiac Anomalies in AAAs

The overall number of intracardiac anomalies within the whole cohort was 36/90 (40.0%). The most affected subgroup was the RAA-mirror subgroup with 100% (13/13), followed by the RAA-ALSA subgroup (10/27; 37.0%) and ARSA subgroup (13/46; 28.3%). No-one within the DAA subgroup (*n* = 4) had an additional intracardiac anomaly. Intracardiac anomalies within the ARSA subgroup (Figure 6) were mainly atrioventricular septal defects (AVSD) or VSD. In the patients with RAA, RAA mirror (Figure 7) and RAA-ALSA (Figure 8), the majority had conotruncal anomalies with tetralogy of Fallot (TOF) as the most common cardiac defect. Intracardiac anomalies within the AAA subgroups ARSA, RAA-mirror and RAA-ALSA are shown in Figure 6, Figure 7 and Figure 8.

### 3.3. Associated Extracardiac Anomalies in AAA

The rate of extracardiac anomalies was comparable in all subgroups with 30.7% in the RAA-mirror, 28.3% in ARSA, 25.0% in DAA- and 22.2% in the RAA-ALSA subgroup (Table 1). A wide spectrum of anomalies was present (Table 2). Severe extracardiac anomalies were often associated with chromosomal anomalies.

### 3.4. Isolated AAAs

Of the whole cohort, 42/90 (46.7%) patients had isolated AAAs without intracardiac or extracardiac anomalies: 26/46 within the ARSA subgroup, 13 within the ALSA subgroup and 3/4 within the DAA subgroup. Of the cases with isolated AAAs, 3/42 (7.1%) had chromosomal anomalies and these cases were all within the ARSA subgroup. There were no patients with isolated AAAs and abnormal karyotype in the other subgroups. The chromosomal anomaly was trisomy 21 in all these (3/26, 11.5%) isolated ARSA cases. One was lost to follow-up (FU) during the prenatal period. The 40-year-old patient was referred for a second trimester sonography to our center after noninvasive prenatal testing (NIPT) with high risk for trisomy 21 and confirmation by amniocentesis in the first trimester and decided to carry out the pregnancy. The other two patients opted for termination of pregnancy (TOP). One case was a 37-year-old patient with referral in 14 + 2 wks for amniocentesis after external increased nuchal translucency and high-risk NIPT for trisomy 21. The examination in our center revealed ARSA. The third 32-year-old patient was referred for a dichorionic twin pregnancy; isolated ARSA in one fetus was diagnosed at 19 + 1 wks in our center. First, the patient opted for NIPT, of which the result showed a 1high risk for trisomy 21. A subsequent amniocentesis revealed trisomy 21 in both fetuses and parents opted for fetocide of both fetuses. An overview of invasive, non-invasive testing and results in patients with isolated AAAs is demonstrated in Table 3.

### 3.5. Prenatal Outcome in AAAs

Of the whole cohort, 19/90 (21.1%) patients with AAAs were lost to FU during the prenatal period. Intrauterine death occurred in 2/90 (2.2%), one in a patient with trisomy 18, AVSD and ARSA at 14 + 5 wks and one in a fetus of monochorionic twin pregnancy with selective fetal growth restriction and RAA-ALSA at 29 + 4 wks. In six (6.7%) cases of our cohort, parents opted for TOP; all six were within the ARSA subgroup and had chromosomal anomalies: one case with trisomy 18, one case with triploidy and four patients with trisomy 21.

### 3.6. Postnatal Outcome in AAAs

Of the whole cohort, 63 (70.0%) of 90 patients were liveborn. In 3 of 63 (4.8%) cases, parents decided for compassionate care, two of them with trisomy 18 died after vaginal delivery at 34 + 3 and 39 + 5 wks and one with TOF, RAA-mirror and extracardiac anomalies (central nervous system malformation, pes equinovarus) died after a caesarean section at 30 + 0 wks. Regarding the perinatal data of the remaining 60 neonates, the median week of gestation at birth was 38.57 wks (25.29–40.23), median birthweight 3200 g (715–4390), median 5 min-APGAR 9 (0–10) and median arterial pH from umbilical cord 7.31 (7.00–7.43). Rate of prematurity < 34 + 0 wks was 4/60 (6.7%) and <37 + 0 wks 10/60 (16.6%). Rate of caesarean section was 31/60 (51.7%), 25/60 (41.7%) had vaginal delivery and in 4/60 (6.7%) the mode of delivery was unknown. Three of sixty (5.0%) patients were lost to FU after birth. Four (7.0%) of the remaining evaluable fifty-seven patients with intention to treat died during FU. One patient of the ARSA subgroup with shone complex and turner died one month postpartum. The three other patients were within the RAA-ALSA subgroup: one patient with RAA-ALSA, microcephaly and anhydramnios since 16 wks died due to early prematurity the day after precipitate delivery at 25 + 2 wks, another patient with TOF+RAA-ALSA and extracardiac anomalies (renal agenesis and meningomyelocele) died after birth and one with truncus arteriosus communis (TAC) with interrupted aortic arch (IAA) of the RAA-ALSA subgroup and late preterm (35 + 4 wks) died 4 days postpartum due to cardiovascular failure as a consequence of dysplastic truncus valve and with severe tricuspid insufficiency. The survival rate was 53/57 (93%) during a median FU of 10.5 months. Twelve (22.6%) of these patients had RAA-mirror and consequently presented without vascular ring/sling. Only 2 (4.9%) of the 41 remaining patients with vascular ring/sling developed symptoms during FU and only one (2.4%) needed an operation. Both were in the RAA-ALSA subgroup. The patient with indications for operation had malalignment VSD and a double-chambered right ventricle as a rare additional intracardiac anomaly. The operation was performed at the age of ten months because of progressive stridor and shallowing problems for at least five months. The other patient with isolated RAA-ALSA had mild stridor and swallowing problems at the age of three months, but symptoms were regressive at a control examination three months later; therefore, there was no indication for an operation until now. None of the four patients within the DAA subgroup had any symptoms during a median FU of 12.5 (range 1–24) months. The outcomes of patients with different AAAs are demonstrated in Figure 9 and outcomes of patients with isolated AAAs cases in Table 3.

Limitations of this study are its retrospective character and the relatively high lost to FU rate, especially during the prenatal period. Moreover, most evaluated cases with referral to our center are high-risk pregnancies; therefore, the incidence of aneuploidy would be higher and therefore associations between genetic anomalies and especially ARSA could be overestimated.

## 4. Discussion

The frequency of AAAs is low with a prevalence of 0.7% for ARSA [9] and only 0.1% for RAA in the postnatal cohort and in a prenatal cohort of low-risk pregnancies [10,11]. The prevalence of fetal double aortic arch is not well known [12,18]. Some estimate an incidence of approximately 0.005–0.007% of fetuses [13,19]. Therefore, data concerning prenatal cases with AAAs, in particular, are limited. The aim of our retrospective evaluation was to evaluate associated anomalies and outcomes, especially operations for vascular ring/sling in prenatally diagnosed cases of ARSA, RAA-mirror, RAA-ALSA and DAA.

### 4.1. AAA Diagnostics of AAA and Accuracy Rate

A correct prenatal ultrasound diagnosis of AAA subtypes by ultrasound was possible in most cases (89%). This is in line with the accuracy rate in the literature with reports in RAA patients ranging from 68% to 100% [6,7,11,25]. In one case, we supposed DAA, but RAA-ALSA was diagnosed after birth; the difficult differentiation during the prenatal period has already been described [25]. In the other ten cases, mainly ARSA cases, diagnosis was not confirmed by the postnatal echocardiography. Yet, in our center these patients did not have further diagnostic imaging like magnetic resonance imaging (MRI). However, some authors suggest neonatal echocardiography as an unreliable diagnostic tool for AAAs [19,26,27], because pulmonary air easily interferes with image quality. Therefore, MRI or computed tomography (CT) is considered to be the gold standard for identifying such AAA variations [19]. In our cohort, a postnatal MRI was mainly performed in patients with additional cardiac defects and/or symptoms of vascular ring/sling. In these patients who had an MRI, prenatally suspected AAAs by fetal echocardiography was confirmed in every case. Moreover, postnatal diagnostics on branching patterns in AAAs are limited and there is often no postnatal systematic verification [6,27]. We also could not include some patients into our analysis because imaging was not performed after birth especially in isolated, clinically inconspicuous ARSA cases. Furthermore, in contrast to a prenatal ultrasound, postnatal echocardiography was not exclusively performed by experienced sonographers, and as a consequence some diagnoses could be missed. In addition, we documented a high rate of lost-to-FU cases (19/90 = 21.1% prenatally; 3/63 = 4.8% postnatally), mainly in cases with isolated ARSA. This could be explained by the fact that these patients had no obligatory indication for further controls by an ultrasound specialist and delivery in a perinatal center with a pediatric heart center was not mandatory. On the other hand, the rate of associated intracardiac anomalies in our whole cohort is quite high at 40%, which is most likely related to a referral of high-risk patients to our center with a focus on fetal echocardiography and associated pediatric heart centers.

### 4.2. ARSA-Isolated versus Non-Isolated Forms

Concerning the ARSA subgroup, the rate of additional intracardiac and/or extracardiac anomalies was high (each 28.3%). In total, 56.5% had isolated ARSA, similar to the findings by others [9,27,28]. Chaoui et al. suggested ARSA as an ultrasound marker for trisomy 21 after preliminary studies [16,29]. However, it is important to differentiate between isolated ARSA and non-isolated ARSA. Some found that isolated ARSA without other intracardiac findings is benign without or only with a weak association to trisomy 21 (Table 4) or microdeletion 22q11.2 [9,17,27,28,30].

They stress the need of differentiating between absolutely isolated ARSA cases and ARSA with other ultrasound anomalies in terms of risk estimation for trisomy 21. The authors conclude that a detailed fetal scan to exclude other anomalies should be assessed and suggest that isolated ARSA alone does not justify routine invasive testing [27,28,30]. As discussed in a review by Scala et al., NIPT could be useful in isolated cases of ARSA without a higher a priori risk for trisomy 21 and after exclusion of other anomalies [33]. However, others found isolated ARSA as an marker for trisomy 21 (Table 4) [31,32]. These inconsistent results are certainly partly due to no uniform definition for isolated ARSA. Depending on the exclusion or inclusion of ultrasound markers or minor anomalies like LPSVC for the isolated ARSA group, the incidence of trisomy 21 differs which makes comparison of the data difficult. In our analysis, the rate of trisomy 21 within the isolated ARSA group was 11.5% (3/26). However, two of these three cases with isolated ARSA occurred in a 37- and 40-year-old patient with high a priori risk for trisomy 21. One of them was referred for planning amniocentesis after high-risk NIPT for trisomy 21 and increased nuchal translucency. In our examination, increased nuchal translucency was not more detectable, but ARSA was diagnosed. Higher values of nuchal translucency as a physiological lymphatic collection are associated with a higher risk of congenital heart disease and trisomy 21. As it is not a malformation and only an ultrasound marker, we defined this case as isolated ARSA. For others, the presence of such markers would perhaps meet criteria for a non-isolated case. The third case occurred in a 32-year-old patient without higher a priori risk for trisomy 21.

Moreover, the incidence of AAAs seems to decrease with ongoing pregnancy and is higher than in the postnatal series [33,34]. As Scala et al., we suppose that this is due to the fact that a relevant number of cases with chromosomal anomalies, including severe forms of trisomy 21, die in utero [33]. The lost-to-FU rate, especially in isolated cases of ARSA, within our cohort was relevant and only less than 20% of our cohort were referred in the first trimester. Therefore, the prevalence of trisomy 21 in isolated ARSA could be over- or underestimated. Moreover, most evaluated cases with referral to our center are high-risk pregnancies with higher maternal age; therefore, the incidence of aneuploidy would be higher and therefore the association between genetic anomalies and ARSA could be overestimated.

Apart from trisomy 21, ARSA is associated with other genetic disorders, especially microdeletion 22q11.2 [24,35]. We documented one case of microdeletion 22q11.2 with TOF and ARSA. However, as pointed out by Pico et al., the exclusion of microdeletion 22q11.2 after clinical examination by neonatologists in an asymptomatic newborn is more difficult than for trisomy 21 [27]. A postmortem evaluation detected 16% of ARSA within a cohort of fetuses with microdeletion 22q11.2 [36]. Only 45.7% of our ARSA cases had karyotyping so missing such a diagnosis is possible.

We think NIPT could be a good option in isolated ARSA cases and should be offered during parents counseling.

### 4.3. Associated Anomalies in RAA

Concerning our subgroup with RAA-mirror, we documented associated cardiac defects in all cases. This strong association of RAA without vascular ring formation and mirror branching to cardiac defects has been reported by others [6,7,37]. In our cohort, cardiac defects within this subgroup were mainly TOF as also reported by Berg et al. [6]. In contrast, RAA-ALSA forming a vascular ring shows a lower association with additional intracardiac anomalies [12] and is often an isolated finding. It may be more frequent in the normal population than generally recognized [25,38]. Regarding chromosomal anomalies in RAA-ALSA, the data are inconsistent. Some only recommend testing for microdeletion 22q11.2 in non-isolated forms with other anomalies [25]. However, a recent study found a prevalence of 5% of microdeletion 22q11.5 in patients with isolated RAA and DAA [34]. Another meta-analysis of a cohort with RAA without intracardiac anomalies showed similar results, with a rate about 5% of chromosomal anomalies in completely isolated RAA cases and almost 10% in cases of an additional extracardiac anomaly [38]. The authors also suggest that fetal RAA with normal intracardiac anatomy is more frequently associated with extracardiac anomalies (15–20%) than with chromosomal abnormalities. This is in line with our results with over 20% extracardiac anomalies within the RAA-ALSA subgroup and not even 4% chromosomal anomalies. We documented no chromosomal anomalies within isolated RAA-ALSA cases and none within the DAA subgroup. The only patient with RAA-ALSA and microdeletion 22 q11.2 had microcephaly as additional extracardiac anomaly. In our opinion, invasive diagnostics should be offered in cases of a prenatal diagnosis of RAA or DAA.

Vigneswaran et al. found a higher median maternal age and higher incidence of conceptions from in vitro fertilization (IVF) of about 5% prevalence in their cohort with isolated RAA and DAA compared to those without [34]. In our cohort, the rate of assisted reproductive technology was 10% within the whole cohort, the rate within the RAA-mirror and DAA subgroup was over 20%, respectively, compared to a rate of about 3% in the general German population [39].

### 4.4. Symptoms of Vascular Ring/Sling in AAA

Forming an incomplete vascular ring, ARSA can cause symptoms like dysphagia, chronic cough and stridor in the postnatal period [2,20,21]. However, in contrast to DAA, complete vascular ring symptoms are less frequent and the need for surgery is uncommon [5]. Consistent to the data of Tuo et al. [5], none of our ARSA patients developed symptoms requiring surgery due to vascular sling during postnatal FU. Concerning the postnatal period of AAA cases with vascular ring, the development of symptoms due to tracheoesophageal compression occurs early in life when ductus arteriosus regresses and ligamentum arteriosus appears [18]. A recent review of 40 prenatal DAA cases revealed that more than half of the patients are completely asymptomatic and only 42% needed surgical correction (Table 5) [19].

Concerning patients with RAA, the rate was lower with 25% for symptoms and 17% indication for operation after a recent meta-analysis (Table 5) [38]. In contrast to these results, in our evaluation not even 5% of patients developed symptoms and only one patient with RAA-ALSA (2.4%) needed an operation during FU. This is in line with the findings of Achiron et al. with also only one patient (5.3%) with vascular ring and indication for surgery. In their cohort, it was a patient with DAA (Table 5) [11]. In contrast to them and others, none of our patients with confirmed DAA developed any pressure symptoms [5,6,11]. However, due to the rareness of incidence of DAA this was the smallest subgroup within our cohort. Symptoms are unlikely to occur in the neonatal period or after 24 months postpartum; therefore, postnatal surveillance is mainly important in the first two years [38]. Our median postnatal FU time was 10.5 months and within the DAA group it was lower than 6 months in two patients. For this reason, the rate of symptoms and resulting rate of operations could be underestimated. On the other hand, results with a similar low rate of operations have been shown in studies with a longer mean FU time of 60 (12–80) months [11].

## 5. Conclusions

After diagnosis of AAAs, a detailed fetal scan including fetal echocardiography should be assessed by a specialist to exclude associated intracardiac and extracardiac anomalies. An invasive diagnostic should be recommended in non-isolated forms. Symptoms due to vascular ring are rare and the majority do not need intervention for that. In isolated ARSA cases, we observed three cases of trisomy 21, but two of them with high a priori risk. We documented no chromosomal abnormalities in isolated RAA or DAA cases. An invasive diagnostic should be restrained in these cases and NIPT could perhaps be an alternative for isolated ARSA cases.

## Figures and Tables

**Figure 1 diagnostics-14-00238-f001:**
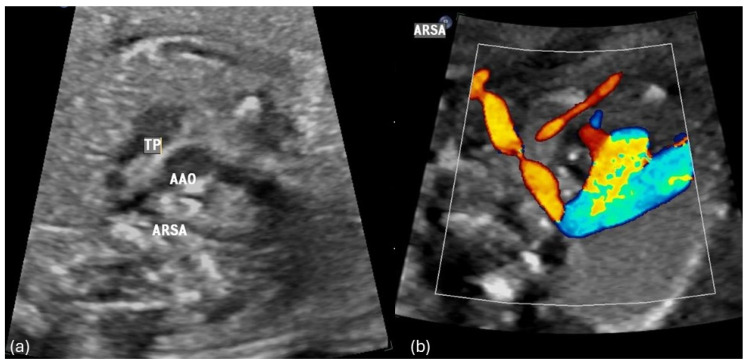
(**a**,**b**) Fetal transverse view in a case of left aortic arch with an aberrant right subclavian artery (ARSA). The ascending aorta (AAO) and the pulmonary trunk (TP) and arterial duct appear to the left of the trachea in the three vessel view forming a V -shaped structure. The right subclavian artery crosses the fetal chest posterior to the trachea toward the right upper limb forming a vascular sling around the trachea and esophagus.

**Figure 2 diagnostics-14-00238-f002:**
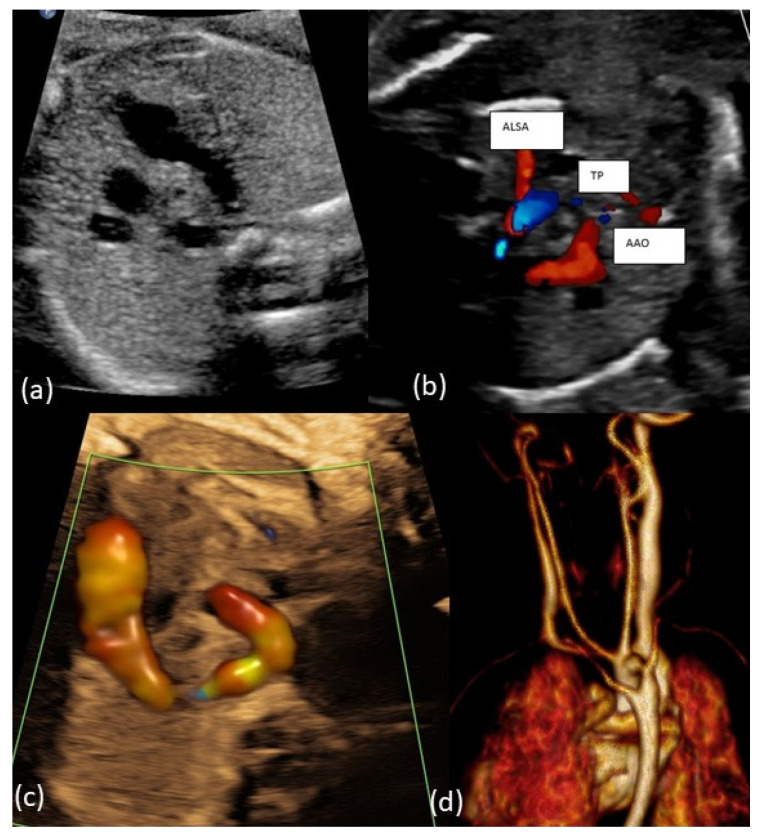
(**a**–**c**): Fetal transverse view in a case with right aortic arch and aberrant left subclavian artery (ALSA). The aortic arch is located to the right of the trachea in the three-vessel view forming a U-shaped structure with the arterial duct being left. The left subclavian artery crosses the fetal chest posterior to the trachea toward the left upper limb forming a vascular ring around the trachea and esophagus. (**d**): Postnatal MRI of newborn with RAA-ALSA.

**Figure 3 diagnostics-14-00238-f003:**
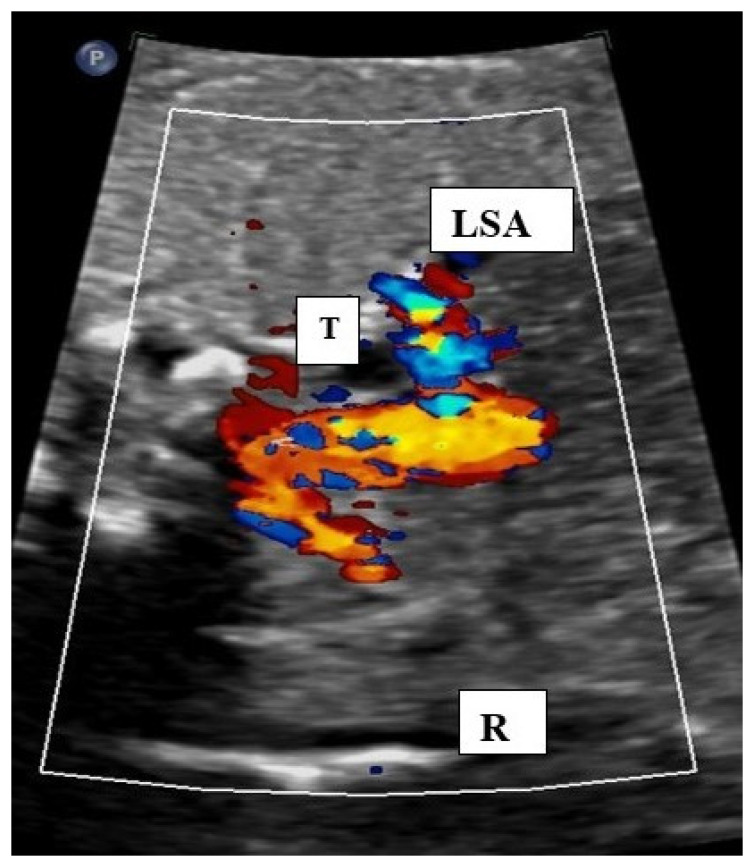
Fetal transverse view in a case of right aortic arch with mirror image branching. The aortic arch is situated on the right (R) of the trachea.. Both subclavian arteries can be seen in an antetracheal position showing a mirror image branching pattern. LSA: left subclavian artery, T: trachea.

**Figure 4 diagnostics-14-00238-f004:**
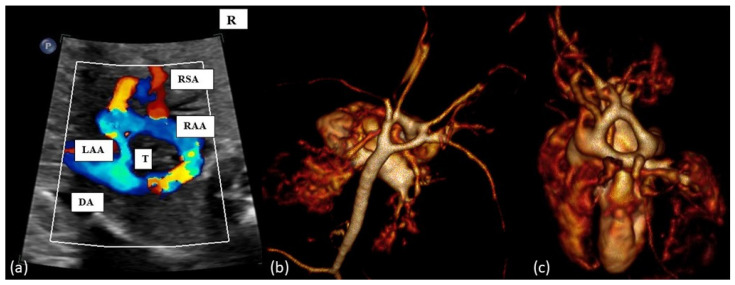
(**a**): Fetal transverse view in a case of double aortic arch. There are four instead of three vessels, with the vessels being right and left aortic arches, the arterial duct (DA, left or right) and the superior vena cava. Typically, the left-sided aortic arch is smaller (LAA) than right-sided aortic arch (RAA) and the left and right subclavian artery show (RSA) an antetracheal course. Double aortic arch forms a tight vascular ring. (**b**,**c**): Postnatal MRI of a newborn with DAA. T = trachea.

**Figure 5 diagnostics-14-00238-f005:**
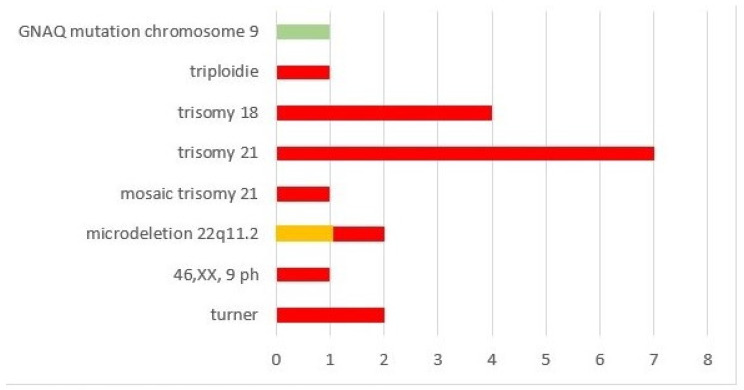
Chromosomal anomalies within our cohort.

**Figure 6 diagnostics-14-00238-f006:**
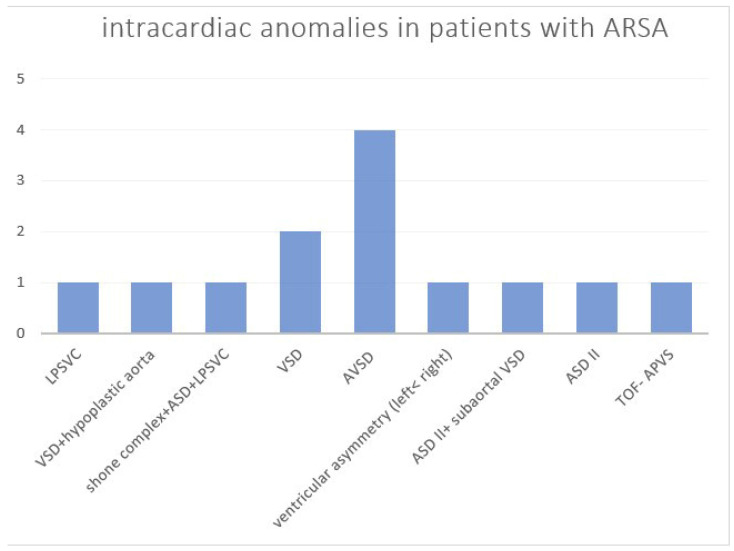
Intracardiac anomalies within the ARSA subgroup. ASD = atrial septal defect; AVSD = atrioventricular septal defect; ARSA = aberrant right subclavian artery; LPSVC = left persistent superior vena cava; TOF-APVS = tetralogy of Fallot with absent pulmonary valve syndrome; VSD = ventricular septal defect.

**Figure 7 diagnostics-14-00238-f007:**
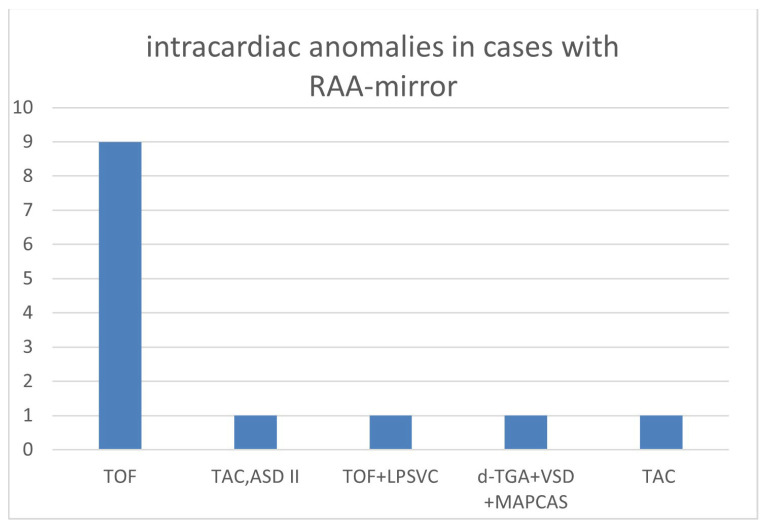
Intracardiac anomalies within right aortic arch with mirror image branching. (RAA-mirror) subgroup; ASD = atrial septal defect; MAPCAS = major aortopulmonary collateral arteries; LPSVC = left persistent superior vena cava; RAA-mirror = right aortic arch with mirror image branching; TAC = truncus arteriosus communis; d-TGA = dextro-transposition of the great arteries; TOF = tetralogy of Fallot.

**Figure 8 diagnostics-14-00238-f008:**
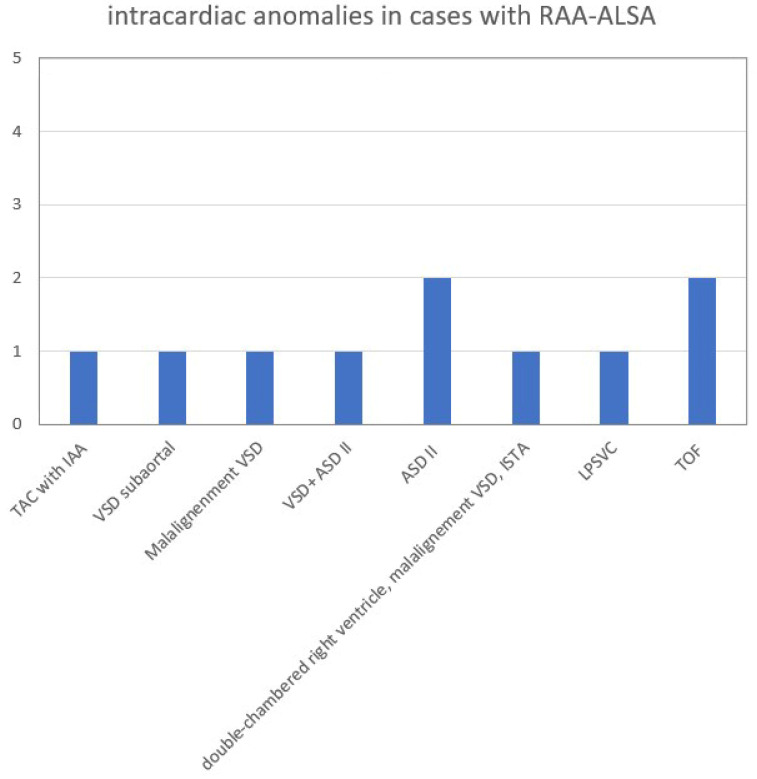
Intracardiac anomalies within RAA-ALSA subgroup. ASD = atrial septal defect; IAA = interrupted aortic arch; ISTA = aortic isthmus stenosis. LPSVC = left persistent superior vena cava; RAA-ALSA = right aortic arch with aberrant left subclavian artery; TAC = truncus arteriosus communis; TOF = tetralogy of Fallot; VSD = ventricular septal defect.

**Figure 9 diagnostics-14-00238-f009:**
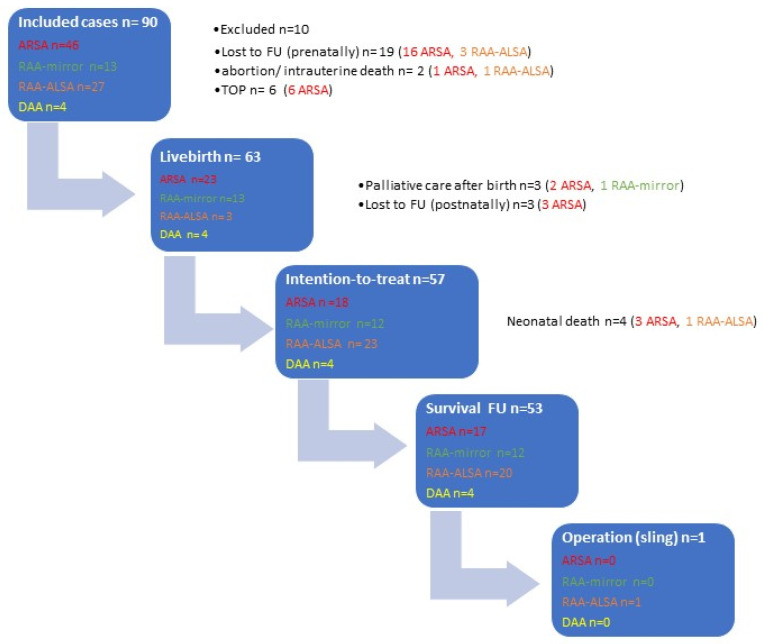
Outcome in our cohort with aortic arch anomalies. ARSA = aberrant right subclavian artery; DAA = double aortic arch; FU = follow-up; RAA-ALSA = right aortic arch with aberrant left subclavian artery; RAA-mirror = right aortic arch with mirror image branching; TOP = termination of pregnancy.

**Table 1 diagnostics-14-00238-t001:** Characteristics and associated anomalies within our cohort.

	All	Twin Pregnancy	ART (IVF/ ICSI)	Genetic Anomalies	NT >95 Percentile	Intracardiac Anomalies	Extracardiac Anomalies	Isolated (No Other Malformation)	Genetic Anomalies within Isolated Cases
ARSA	46	5 (10.9%)	3 (6.5%)	17 (37.0%)	9 (19.6%)	13 (28.3%)	13 (28.3%)	26 (56.5%)	3/26 (11.5%)
RAA-mirror	13	1 (7.7%)	3 (23.1%)	1 (7.7%)	0 (0%)	13 (100%)	4 (30.7%)	0 (0%)	0 (0%)
RAA-ALSA	27	2 (7.4%)	2 (7.4%)	1 (3,7%)	0 (0%)	10 (37.0%)	6 (22.2%)	13 (48.1%)	0 (0%)
DAA	4	0 (0%)	1 (25%)	0 (0%)	0 (0%)	0 (0%)	1 (25%)	3 (75%)	0 (0%)
all	90	8 (8.9%)	9 (10.0%)	19 (21.1%)	9 (10%)	36 (40.0%)	24 (26.7%)	42 (46.7)	3/42 (7.1%)

ARSA = aberrant right subclavian artery; ART = assisted reproductive technology; DAA = double aortic arch; IVF = in vitro fertilization; ICSI = intracytoplasmic sperm injection; NT = nuchal translucency; RAA-ALSA = right aortic arch with aberrant left subclavian artery; RAA-mirror = right aortic arch with mirror image branching.

**Table 2 diagnostics-14-00238-t002:** Extracardiac anomalies within our cohort.

ARSA 13 Patients	RAA-Mirror 4 Patients	RAA-ALSA 6 Patients	DAA 1 Patient
pes eqinovarus (T21) plexus cyst n = 2 hemivertebra megacisterna magna, hypoplastic cerebellar vermis, cleft palate (T18) omphalocele, esophageal atresia, hand deformity, pes equinovarus (T18) single umbilical artery, hydrops fetalis (T18) single umbilical artery double kidney, edema (turner) microcephaly (T21) hypospadia, preauricular appendage duodenal atresia (mosaic T21) appendage right hand, muscular gap proximal to umbilicus (microdeletion 22q11.2)	equinovarus, hydrocephalus, macrocephaly, cerebellar hypoplasia microtia, glaucoma, nevus flammeus (face), angioma (supratentorial) (GNAQ chromosome 9 mutation) hydronephrosis, megaureter, dysplastic ear left esophageal atresia, renal agenesis, Madelung’s deformity left hand, anal atresia with rectovaginal fistula (VACTERL)	syndactyly left hand (III/IV), feet deformity cleft lip radial polydactyly, preauricular fistula, preauricular appendage microcephaly n = 2 (one case with microdeletion 22q11.2) meningomyelocele + renal agenesis	hypospadia

**Table 3 diagnostics-14-00238-t003:** Invasive testing, noninvasive prenatal testing (NIPT) and outcomes within isolated cases of aortic arch anomalies in our cohort.

	*n*	Invasive Testing	NIPT (Without Further Invasive Testing)	Abnormal Result	Outcome in Abnormal Results	Outcome in Patients with Normal Karyotype/Normal NIPT	Outcome Patients without Prenatal Testing/Invasive Testing
ARSA	26	6/26 (23.1%) 3 of them with high-risk NIPT for T21 and subsequent amniocentesis	7	3/26 (11.5%)	1 lost to FU prenatally, 2 TOP	5 lost to FU prenatally 3 lost to FU postpartum 2 livebirths with FU	5 lost to FU prenatally; 8 livebirths with FU
RAA-mirror	0	-	-	-	-	-	-
RAA-ALSA	13	2/13 (15.4%)	2 (Both without testing for microdeletion)	0	-	4 livebirths with FU	6 livebirths 1 IUD 29 + 4 wks in a monochorionic twin with sFGR 2 lost to FU prenatally
DAA	3	0/3 (0%)	0	0	-	-	3 livebirths, no operation during FU

ARSA = aberrant right subclavian artery; DAA = double aortic arch; FU = follow-up; IUD = intrauterine death; sFGR = selective fetal growth restriction; RAA-ALSA = right aortic arch with aberrant left subclavian artery; RAA-mirror = right aortic arch with mirror image branching; TOP = termination of pregnancy; wks = weeks.

**Table 4 diagnostics-14-00238-t004:** Data from different studies related to ARSA/isolated ARSA and association with chromosomal anomalies.

Reference	Cohort	ARSA (Total)	ARSA (Total) with Chromosomal Anomalies	Isolated ARSA	Isolated ARSA with Chromosomal Anomalies
Chaoui et al., 2005 [16]	14 fetuses with trisomy 21 (HR)	5	5	1	1
Gul et al., 2011 [31]	4125 fetuses (LR)	17	1	9	1
Paladini et al., 2012 [32]	106 fetuses with trisomy 21 (HR)	27	27	8	8
Pico et al., 2016 [27]	120 fetuses with ARSA (108 with outcome) (M)	108	22 (9 with T 21)	54	0
Ranzini et al., 2017 [28]	79 fetuses with ARSA (M)	79	11 (7 with T 21)	43	0
Willruth et al., 2012 [30]	1337 fetuses (M)	14	3 (1 × T21)	9	0
Zalel et al., 2008 [17]	924 fetuses (M)	16	3 (3 × T21)	6	0
our results	46 fetuses with ARSA (M)	46	17 (7 × T21)	26	3

HR = high-risk population; LR = low-risk population; M: mixed population.

**Table 5 diagnostics-14-00238-t005:** Operation rate for vascular ring/sling in the literature compared with our results.

Reference	Cohort	FU Time (Months)	Operation (Vascular Ring/Sling)
Achiron et al., 2002 [11]	19 fetuses with vascular ring/sling	60 (median)	1/19 (5.3%) (DAA case)
Guo et al., 2020 [19]	40 fetuses with DAA, 27 live births with DAA	38 ± 17 (mean)	11/27 (41%)
Tuo et al., 2009 [5]	19 fetuses with vascular ring/sling	23.4 (mean)	4/19 (21.1%) (3 × DAA, 1 × RAA-ALSA)
Berg et al., 2006 [6]	71 fetuses, 28 with vascular ring/sling	minimum 12 for each case	1/25 (3.6%) (DAA case)
Our results	41 livebirths with vascular ring	10.5 (median)	1/41 (2.4%) (RAA-ALSA case)

DAA = double aortic arch; RAA-ALSA = right aortic arch with aberrant left subclavian artery; FU = follow-up.

## Data Availability

Data are unavailable due to ethical restrictions.

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
