# Peer review of "Associated Anomalies and Outcome in Patients with Prenatal Diagnosis of Aortic Arch Anomalies as Aberrant Right Subclavian Artery, Right Aortic Arch and Double Aortic Arch"

_diagnostics, 2024, doi:10.3390/diagnostics14030238_

Round 1

Reviewer 1 Report

Comments and Suggestions for Authors

Wolter et al evaluated retrospectively associated anomalies and outcome in prenatal aortic  arch anomalies. This manuscript is well-written. The reviewer had several comments as follows:

1.     Please provide the expansion for NIPT in Abstract section.

2.     Examples of of aortic arch anomalies (Figures 1-4) could be mentioned in Methods section, rather than Introduction section.

3.     Figures 1a and 1b should be merged into one composite FIGURE. Similarly, Figrues 2a-d and 4a-c should be revised.

4.      Discussion: This section of the manuscript would benefit from subheadings.

5.     The limitations of this study should be mentioned.

Comments on the Quality of English Language

This manuscript is well-written. It has no spelling errors or grammatical errors.

Author Response

  1. Please provide the expansion for NIPT in Abstract section.

Thank you for your comment. We provide the expansion for NIPT within abstract section more precisely: “NIPT could be useful in isolated ARSA-cases without higher a-priori-risk for trisomy 21 and after exclusion of other anomalies.” The abstract section is limited to 200 words therefore more detailed formulation in abstract section is not possible.

  1. Examples of aortic arch anomalies (Figures 1-4) could be mentioned in Methods section, rather than Introduction section.

Thank you for your comment. We mentioned it in Methods section.

  1. Figures 1a and 1b should be merged into one composite FIGURE. Similarly, Figures 2a-d and 4a-c should be revised.

Thank you for your comment. We merged figure 1a and figure 1b into one figure as well as figure 2a-d and figure 4 a-c.

  1. Discussion: This section of the manuscript would benefit from subheadings.

Thank you for your comment. We added subheadings within the discussion-part.

  1. The limitations of this study should be mentioned.

Thank you for your comment. We added limitations of the study in the conclusion part: Limitation of this study are its retrospective character, the relatively high lost to FU rate especially during prental period. Moreover most evaluated cases with referral to our center are high-risk pregnancies, therefore the incidence of aneuploidy would be higher and therefore association between genetic anomalies and especially ARSA would be overestimated.

Reviewer 2 Report

Comments and Suggestions for Authors

Overall, this is a well written study on prenatal diagnosis of aortic arch anomalies and associated genetic and extracardiac anomalies. The total number of cases with adequate prenatal and postnatal follow-up is somewhat limited.

In addition, a more expansive background review of the literature of this topic should be included- ie incidence of aberrant R subclavian artery in low risk fetuses and neonates.

The authors should also include in their discussion how a diagnosis of aortic arch anomalies in the fetus can affect family counseling prenatally.

The introduction should include a discussion of other studies on arch anomalies and associated genetic and extracardiac anomalies. There are studies that report increased incidence of trisomy 21 with aberrant R subclavian artery, however the incidence of this is in the general low risk population is not known.

Please correct English grammatical errors:  Abstract- "lost to follow up" rather than " lost for follow-up". There is also inconsistency with decimals- should use "." rather than "," in tables in particular.

In methods or results the authors should better describe the study population group- was this a center that performed general screening for low risk population or higher risk groups with fetal echo?

I would suspect that if this was a high-risk referral center, the incidence of aneuploidy would be higher and therefore the association between genetic abnormalities and aberrant right subclavian artery would be overestimated. This should be discussed further in the discussion.

Comments on the Quality of English Language

There are minor grammatical errors. For decimals, in English we typically use "." rather than "," and this is inconsistently done (going back and forth) throughout the paper and tables- please review the entire paper and tables and keep consistent.

In the Results section 3.2 - there is likely an error on line 245: "Intracardiac anomalies with ALSA subgroup were " where the authors likely meant ARSA- please clarify.

Author Response

Overall, this is a well written study on prenatal diagnosis of aortic arch anomalies and associated genetic and extracardiac anomalies. The total number of cases with adequate prenatal and postnatal follow-up is somewhat limited.

In addition, a more expansive background review of the literature of this topic should be included- ie incidence of aberrant R subclavian artery in low-risk fetuses and neonates.

Thank you for your comment. We included a more expansive background of the literature especially concerning incidence of ARSA in the introduction-part:

Although most AAA are asymptomatic during postnatal life, during prenatal period they could be a marker for chromosomal anomalies and a detailed scan for further anomalies should be performed and it should be considered in prenatal family counseling [12].

AAA are associated with genetic and further cardiac and extracardia anomalies. In postnatal period ARSA is present in 3 % of children and adults with congenital herat defect and in 0.1%without heart defect [13]. After postnatal radiographic studies ARSA occurs  often in cases with triomy 21 in up to 35% [14,15]. Chaoui et al suggested ARSA as ultrasound marker for trisomy 21 after preliminary studies. In a prenatal cohort with trisomy 21 they found ARSA in in 35.7 %[16]. Data concerning incidence of ARSA in low risk fetuses is rare. Zalel et al reported an incidence of 1.4 % of ARSA in a fetal cohort with routine sonographic examination. [17]

The authors should also include in their discussion how a diagnosis of aortic arch anomalies in the fetus can affect family counseling prenatally.

Thank you for your comment. We included it into discussion (part 4.2. and 4.3).

The introduction should include a discussion of other studies on arch anomalies and associated genetic and extracardiac anomalies. There are studies that report increased incidence of trisomy 21 with aberrant R subclavian artery, however the incidence of this is in the general low risk population is not known.

Thank you for your comment. We included it into the introduction part.

Please correct English grammatical errors:  Abstract- "lost to follow up" rather than " lost for follow-up". There is also inconsistency with decimals- should use "." rather than "," in tables in particular.

Thank you for your comment. We corrected it into “ lost to follow up” and used “.” instead of “,”.

In methods or results the authors should better describe the study population group- was this a center that performed general screening for low-risk population or higher risk groups with fetal echo?

Thank you for your comment. We described the study population in methods part:

”In our centre is examinated a mixed population rather higher risk population with some cases for general screening and many who are referred for fetal echocardiography.”

I would suspect that if this was a high-risk referral center, the incidence of aneuploidy would be higher and therefore the association between genetic abnormalities and aberrant right subclavian artery would be overestimated. This should be discussed further in the discussion.

Thank you for your comment. We included this point of discussion in part 4.2. and mentioned it also in the conclusion part limitations of the study.

Comments on the Quality of English Language

There are minor grammatical errors. For decimals, in English we typically use "." rather than "," and this is inconsistently done (going back and forth) throughout the paper and tables- please review the entire paper and tables and keep consistent.

Thank you for your comment. We corrected into “.” instead of “,”.

In the Results section 3.2 - there is likely an error on line 245: "Intracardiac anomalies with ALSA subgroup were " where the authors likely meant ARSA- please clarify.

Thank you for your comment. We corrected it into ARSA subgroup.

Round 2

Reviewer 1 Report

Comments and Suggestions for Authors This revision of the manuscript improved. I have one comment. The limitations of the study should be moved before Discussion section. Comments on the Quality of English Language

Minor editing of English language required

Author Response

Thank you for your comment.  We moved the limitation of the study part before discussion section.

We edited English language in the following paragraphs into:

                                                    Introduction part:

Line 55 ff: Although most AAA are asymptomatic during postnatal life, prenatally they could be a marker for chromosomal anomalies, therefore a detailed scan for further anomalies is necessary and it should be considered in prenatal family counseling [12].

AAA are associated with genetic and further cardiac and extracardiac anomalies. In postnatal period ARSA is present in 3 % of children and adults with congenital herat defect versus in 0.1%without heart defect [13]. After postnatal radiographic studies ARSA occurs in cases with trisomy 21 in up to 35% [14,15]. Chaoui et al suggested ARSA as ultrasound marker for trisomy 21 after preliminary studies. In a prenatal cohort with trisomy 21 they found ARSA in in 35.7 %[16]. Data concerning incidence of ARSA in low risk fetuses is rare. Zalel et al reported an incidence of 1.4 % of ARSA in a fetal cohort with routine sonographic examination. [17]

Line 67: Some estimates an incidence of approximately 0.005% ~ 0.007% [19].

Patients and Methods part:

Line 88: In our centre a rather high-risk population presents with a low number of cases for general screening and most cases with referral for detailed fetal echocardiography.

Line 100: Informed parental consent to anonymized analysis of the data was obtained.

Results part:

Line 259/260: Patients were referred to our centre at 22.1 wks (11.43-30.43) in median. 15/90 (16.7%) presented within the first trimester.

Line 294: No one within DAA subgroup (n=4) had additional intracardiac anomaly.

Line 398: . Subsequent amniocentesis revealed trisomy 21 in both fetuses and parents opted for fetocide of both fetuses.

Line 424: In six (6.7%) cases of our cohort parents opted for TOP, all six were within ARSA-subgroup and had chromosomal anomalies: one case with trisomy 18, one case with triploidy and four patients with trisomy 21.

Line 431: In three of 63 (4.8%) cases parents decided for compassionate care, two of them with trisomy 18 died after vaginal delivery in 34+3 and 39+5 wks, one with TOF, RAA-mirror and extracardiac anomalies (central nervous system malformation, pes equinovarus) died after ceasarean section in 30+0 wks.

Line 460: None of four patients within DAA subgroup had any symptoms during median FU of 12.5 (range 1-24) months in this subgroup.

Line 462: Outcome of patients with different AAA is demonstrated in figure 9 and outcome of patients with isolated AAA cases in table 3.

Line 481: Limitation of this study are its retrospective character and  the relatively high lost to FU rate especially during prental period. Moreover, most evaluated cases with referral to our center are high-risk pregnancies, therefore the incidence of aneuploidy would be higher and therefore association between genetic anomalies and especially ARSA could be overestimated.

Discussion part:

Line 499:

In one case we supposed DAA, but RAA-ALSA was diagnosed after birth, the difficult differentiation during prenatal period has already been described [25].

Line 502: But in our centre these patients did not have further diagnostic imaging like magnetic resonance imaging (MRI).

Line 511: We also could not include some patients into our analysis because imaging was not performed after birth especially in isolated, clinically inconspicuous ARSA cases.

Line 513: . Furthermore, in contrast to prenatal ultrasound, postnatal echocardiography was not exclusively done by experienced sonographers, as a consequence some diagnoses could be missed.

Line 520: On the other hand, the rate of associated intracardiac anomalies in our whole cohort is quite high with 40% which is most likely related to referral of high-risk patients to our centre with focus on fetal echocardiography and associated pediatric heart centre.

Line 530: Some found that isolated ARSA without other intracardiac findings is benign without or only weak association to trisomy 21 (table 4) or microdeletion 22q11.2. [9,17,27,28,30].

Line 540: They stress the need of differentiating between absolutely isolated ARSA-cases and ARSA with other ultrasound anomalies in terms of risk estimation for trisomy 21.

Line 555: In our examination increased nuchal translucency was not more detectable, but ARSA was diagnosed.

Line 566: The lost to FU rate especially in isolated cases of ARSA within our cohort was relevant and only less than 20% of our cohort were referred in first trimester. Therefore, prevalence of trisomy 21 in isolated ARSA could be over- or underestimated. Moreover, most evaluated cases with referral to our center are high-risk pregnancies with higher maternal age, therefore the incidence of aneuploidy would be higher and therefore association between genetic anomalies and ARSA could be overestimated.

Line 590: It may be more frequent in the normal population than generally recognized [25,39].

Line 615: Consistent to data of Tuo et al [5], no one  of our ARSA patients developed symptoms requiring surgery due to vascular sling during postnatal FU.

Line 645: For this reason, rate of symptoms and resulting rate of operations could be underestimated.

Line 646: On the other hand, results with similar low rate of operations have been shown in studies with  longer mean FU time of 60 (12-80) months [11].